# Inverse problems for dynamic patterns in coupled oscillator networks: when larger networks are simpler

Oleh E. Omel'chenko ✉

Networks of coupled phase oscillators are one of the most studied dynamical systems with numerous applications in physics, chemistry, biology, and engineering. Their behaviour is often characterized by the emergence of various partially synchronized dynamic patterns, which in the case of large networks can be analysed using a variant of the mean-field approach. This method allows to predict what type of network dynamics can be observed for different system parameters. But it is less known that for different partially synchronized patterns it also allows to obtain statistical equilibrium relations that express the dependence of some time-averaged observable quantities of individual oscillators on the internal parameters of these oscillators and the interaction functions between them. In this paper, we show how such relations can be derived, what their typical accuracy is for finite-size networks, and how they can be used to reconstruct the parameters of the corresponding model. The proposed method is particularly effective for large networks, for unevenly sampled or noisy observables, and for partial observations. Its possibilities are demonstrated by application to chimera states in networks of phase oscillator with nonlocal coupling. The extension of the method to other systems with all-to-all and random network topologies is also described.

One of the main goals of natural sciences is to predict the behaviour of a given system, assuming that changes in its state are determined by certain dynamical rules expressed by differential equations. In some cases, these equations can be derived from first principles and the results of specially designed experiments, but more often they have to be obtained from uncontrolled observational data. This duality is reflected in the coexistence of two general approaches to the identification of dynamical systems: model-based and data-driven. At first glance, the latter approach seems to be more versatile, as it relies on a minimal amount of information about the system, such as the assumption of sparsity of the governing equations[1]. However, its implementation typically requires a large amount of data (e.g., many trajectories passing through different parts of the phase space) and can become computationally cumbersome as the system size increases. To overcome these difficulties, a number of more sophisticated methods have been proposed, including an equation-free method for inferring coarse-grained multiscale dynamics[2], automated adaptive model inference[3], data-driven discovery of intrinsic lower-dimensional dynamics[4,5], machine learning techniques based on reduced order models[6], and others[7]. A common feature of all these methods is that they attempt to approximate the behaviour of a complex large-scale system using a phenomenological lower-dimensional model, although they utilize this simplification ansatz almost heuristically.

A similar dimensionality reduction scheme also exists in the model-based approach. But there it is better justified and can be used more effectively and purposefully. Roughly speaking, it is well-known that large systems of many interacting agents have the property of coordinating their behaviour in such a way that it is described by the laws of statistical physics. This means that no matter how complex the microscopic dynamics of the system is, it is characterized by a certain statistical balance between the dynamics of the constituent agents and their intrinsic properties. Usually, this relationship is described at the

Institute of Physics and Astronomy, University of Potsdam, Potsdam, Germany. ✉e-mail: omelchenko@uni-potsdam.de

macroscopic level using global coarse-grained variables and some form of mean-field analysis, while the detailed balance at the microscopic level remains in the shadows. In this paper, we show that mathematical formulas expressing this detailed balance can actually be very useful, in particular, for reconstructing the parameters of the corresponding high-dimensional dynamical systems. The general scheme of the proposed approach is described in the context of its application to complex dynamic patterns in networks of coupled phase oscillators. Using it, we formulate a parameter reconstruction algorithm that is non-invasive, fast, easy to compute, suitable for partial observation and robust to measurement noise.

Mathematical models describing the collective behaviour of large populations of coupled phase oscillators can be found in various fields of physics, chemistry, and biology[8,9]. They play a key role in the study of synchronization phenomena[10–12] and have a direct connection to more complex real-world models through the standard phase reduction procedure[10,13–15]. Even without a rigorous justification from first principles, such models are often used in theoretical biology and neuroscience to explain observed properties of dynamical quorum sensing[16,17], circadian rhythm generators[18,19], metachronal waves in cilia carpets[20], brain disorders[21,22], and other physiological processes related to synchrony[23]. In general, these models are defined as follows. The population consists of $N$ oscillators, the state of each of which is described by a scalar quantity, its phase $\theta_j$. Each oscillator has a label $p_j \in \mathbb{R}^{N_p}$ containing information about its intrinsic properties (e.g., natural frequency, position in space, etc.), which remain unchanged over time. Accordingly, the dynamics of this oscillator is determined by a differential equation

$$\frac{d\theta_j}{dt} = F(\theta_j, p_j, W_j),\tag{1}$$

where

$$W_j = \frac{1}{N}\sum_{k=1}^{N} Q(p_j, p_k, \theta_k)\tag{2}$$

is a mean-field acting on the $j$th oscillator due to the influence of all other oscillators. Note that, despite their simple structure, Eqs. (1), (2) describe a broad class of coupled oscillator networks, including fully connected and spatially extended networks, as well as annealed approximations of random networks.

In the thermodynamic limit, when the number of oscillators $N$ tends to infinity and the distribution of labels $p_j$ converges to some probability density $g(p)$, it is often observed that after a sufficiently long transient, the state of system (1) approaches some statistical equilibrium. In the mean-field approximation, this equilibrium is characterized by a single particle probability density function $\rho(\theta, p, t)$. Using this function, we can replace sum (2) with the integral

$$W_j \mapsto \mathcal{W}[\rho](p_j) = \int_{\mathbb{R}^{N_p}} \int_{-\pi}^{\pi} Q(p_j, p, \theta)\rho(\theta, p, t)d\theta\, dp$$

and write a nonlinear integro-differential continuity equation

$$\frac{\partial \rho}{\partial t} + \frac{\partial}{\partial \theta}(F(\theta, p, \mathcal{W}[\rho])\rho) = 0\tag{3}$$

which describes the evolution of $\rho(\theta, p, t)$.

Although Eq. (3) looks more complicated than the original oscillator system (1), its solution representing the statistical equilibrium of (1) has usually a much simpler form than the corresponding oscillators' trajectory. In many cases, this solution $\rho_{\text{se}}(\theta, p, t)$ can be written in analytical (but not necessarily explicit) form, using some kind of self-consistency analysis[10]. Then due to the ergodicity property of

statistical equilibrium the solution $\rho_{\text{se}}(\theta, p, t)$ can be used to derive statistical equilibrium relations, i.e. formulas relating the time-averaged observables in system (1) and the parameters of this system. For example, one of the most common quantities characterizing the dynamics of the $j$th oscillator is its effective frequency, which is defined as

$$\Omega_j = \left\langle \frac{d\theta_j}{dt} \right\rangle,$$

where $\langle \cdot \rangle$ denotes time average. Using Eq. (1), the same value can also be written as

$$\Omega_j = \left\langle \int_{-\pi}^{\pi} \frac{\rho_{\text{se}}(\theta, p_j, t)}{g(p_j)} F(\theta, p_j, \mathcal{W}[\rho](p_j))d\theta \right\rangle,\tag{4}$$

where the use of the conditional probability density $\rho_{\text{se}}(\theta, p_j, t)/g(p_j)$, is due to the fact that we are considering an oscillator with label $p_j$. Formula (4) gives an algebraic relationship between the time-averaged observable $\Omega_j$ and the system parameters $\{p_j\}$. In other words, it expresses the microscopic balance between the dynamics of individual oscillators and their intrinsic properties, and can therefore be considered as a statistical equilibrium relation.

Similar relations, but for other time-averaged quantities, will be described below. In addition, we will show how they can be used to reconstruct the parameters of model (1), (2). For clarity, we will focus on a special but important case − the Kuramoto-Battogtokh system of nonlocally coupled phase oscillators[24]. It is famous as a prototype system for chimera states[25–27], which are dynamic patterns with self-organized domains of synchronized (coherent) and desynchronized (incoherent) behaviour.

## Results

The structure of this section is graphically presented in Fig. 1. First, we describe the Kuramoto-Battogtokh system and show a typical example of a chimera state. Then, we define additional time-averaged quantities, the local order parameters, and write down statistical equilibrium relations for them. (The mathematical details of their derivation can be found in the section Methods.) Finally, we describe our parameter reconstruction algorithm and demonstrate its effectiveness on various examples.

### Model

We consider a ring of $N$ nonlocally coupled identical phase oscillators

$$\frac{d\theta_j}{dt} = \omega - \frac{2\pi}{N}\sum_{k=1}^{N} G(x_j - x_k)\sin(\theta_j - \theta_k + \alpha).\tag{5}$$

Here, $\omega$ is the natural frequency of all oscillators and $\alpha$ is the Kuramoto-Sakaguchi phase lag parameter. The position of the $j$th oscillator is given by $x_j \in [-\pi, \pi]$ and the nonlocal interaction between oscillators is determined by a scalar symmetric $2\pi$-periodic coupling function $G(x)$. More precisely, the positions $x_j$ are assumed to be uniformly distributed on the interval $[-\pi, \pi]$, although in most numerical examples below we will use a special deterministic choice $x_j = -\pi + 2\pi j/N$.

### Pattern

It is well-known[24,25] that for a wide range of parameters in (5) this system exhibits peculiar spatio-temporal patterns, where some oscillators rotate almost synchronously, while others exhibit mutually asynchronous behaviour, see Fig. 2a. In the literature, they are usually called coherence-incoherence patterns or chimera states. Nonlocal couplings for which chimera states have been found include

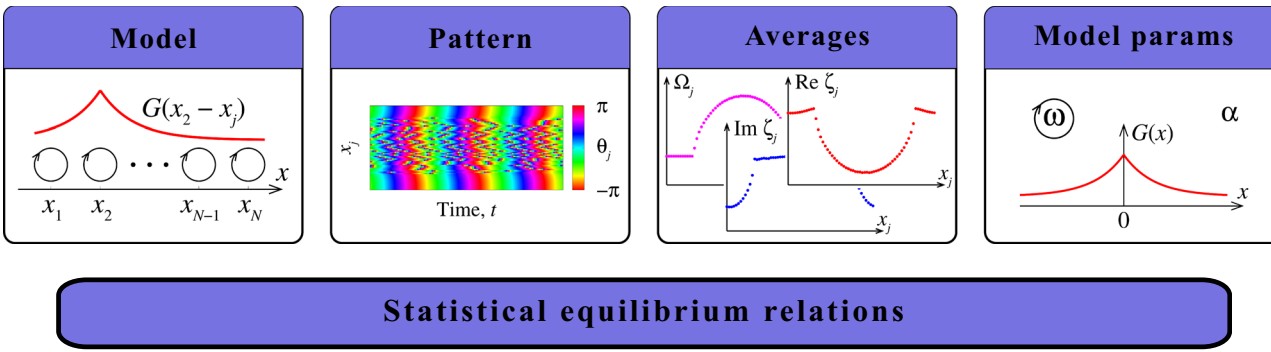

**Fig. 1 | Schematic representation of the proposed parameter reconstruction method.** Given a complex spatio-temporal pattern in a system of coupled phase oscillators, the model parameters can be reconstructed by calculating a small number of averages and using statistical equilibrium relations relevant to this model.

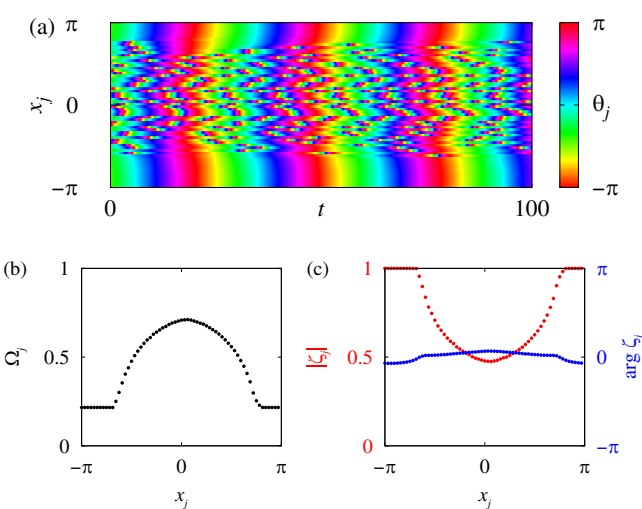

**Fig. 2 | Time-averaged characteristics of chimera states.** A typical chimera state in system (5) for a top-hat coupling function with $\sigma = 0.7$, $\omega = 1$, $\alpha = \pi/2 - 0.1$, and $N = 1024$. **a** Space-time plot of $\theta_j(t)$. **b, c** Effective frequencies $\Omega_j$ and local order parameters $\zeta_j$ obtained by averaging over 2000 time units. Every 16th point $x_j$ is shown.

exponential function[24]

$$G(x) = \frac{\kappa}{2(1 - e^{-\pi\kappa})} e^{-\kappa \arccos(\cos x)}, \ \kappa > 0,$$

cosine function[25]

$$G(x) = \frac{1}{2\pi}(1 + A\cos x), \ A > 0,$$

top-hat function[28]

$$G(x) = \frac{1}{4\pi\sigma}\left(1 + \frac{\pi\sigma - \arccos(\cos x)}{|\pi\sigma - \arccos(\cos x)|}\right), \ 0 < \sigma < 1,$$

and many others[27]. (Note that due to the periodic boundary conditions in model (5), above we used the expression arccos(cos x), which is equal to |x| if |x|≤π, and defines a 2π-periodic extension of |x| if |x| > π.)

The initial interest in chimera states was purely theoretical. But later their existence was confirmed experimentally in systems of chemical Belousov-Zhabotinsky oscillators[29,30] and in systems of electrochemical oscillators[31]. In addition, their similarity to dynamic patterns in various biological systems has been established. These include synchronization patterns of elastic cilia[32,33], collective states of coupled

inner-ear hair cells[34], and epileptic seizures[35]. Although the functional role of chimera states remains unclear, one could consider using them to obtain information about the chemical or biological system in which they occur. For example, the function $G(x)$ in Eq. (5) contains important characteristics of the nonlocal coupling, such as its range, its monotonic (or not) dependence on distance, and its decay rate. The phase lag $\alpha$ is a measure of the nonreciprocity of the interaction between oscillators[36], while the natural frequency $\omega$ is related to the properties of the oscillators in isolation. So, what can we do to find all these interesting parameters in a situation where they cannot be measured directly? More specifically, we can ask the following questions. (i) Can these parameters be determined from the observation of a single chimera state in system (5)? (ii) And if so, how can this be done effectively? Below we will give an affirmative answer to the first question and propose a relatively simple algorithm for solving the second question.

At first glance, the following approach seems to be the most natural to address the parameter reconstruction problem for system (5). Insert the observed trajectory $\{\theta_j(t)\}$ and its derivative $\left\{\frac{d\theta_j}{dt}(t)\right\}$ into Eq. (5) and solve the resulting system with respect to the unknown parameters[37]. However, this method has a number of disadvantages. First, its implementation requires knowledge of the trajectory with high time resolution for accurate calculation of derivatives. Second, the calculations use the entire trajectory $\{\theta_j(t)\}$ as a rectangular matrix, which becomes extremely huge for large system sizes $N$. Third, the trajectory of system (5) must be complete, that is, the behaviour of all oscillators must be known.

## Averages

Below we describe an alternative parameter reconstruction method that does not have the above drawbacks. It is based on statistical equilibrium relations for system (5) and only requires computing $O(N)$ time-averaged quantities from the trajectory $\{\theta_j(t)\}$. Knowledge of derivatives is not required at all. More precisely, for each oscillator $\theta_j(t)$, we only need to calculate its effective frequency $\Omega_j$ and its local order parameter

$$\zeta_j = \left\langle e^{i\theta_j(t)} \frac{\overline{Z}(t)}{|Z(t)|} \right\rangle \in \mathbb{C},$$

where

$$Z(t) = \frac{1}{N}\sum_{k=1}^{N} e^{i\theta_k(t)} \tag{6}$$

is the global order parameter of all oscillators and $\overline{Z}(t)$ is its complex conjugate value, see Fig. 2b, c. Note that after calculating $\Omega_j$ and $\zeta_j$, the oscillator trajectory $\{\theta_j(t)\}$ is no longer needed and does not need to be stored.

## Statistical equilibrium relations (SER)

In the thermodynamic limit, chimera states have an analytic representation following from the corresponding continuity equation (3). Using it, we can derive statistical equilibrium relations (see Methods)

$$\frac{\omega - \Omega_j}{\omega - \Omega} = \frac{2|\zeta_j|^2}{1+|\zeta_j|^2}, \tag{7}$$

$$\mathrm{Re}\left(\frac{\xi_j}{\zeta_j}\right) = \frac{2}{1+|\zeta_j|^2}, \tag{8}$$

$$\xi_j = \frac{2\zeta_j}{1+|\zeta_j|^2} \quad \text{for } |\zeta_j| < 1, \tag{9}$$

where

$$\xi_j = \frac{e^{i\beta}}{\omega - \Omega}\sum_{k=1}^{N} G(x_j - x_k)\zeta_k \frac{x_{k+1} - x_{k-1}}{2} \tag{10}$$

and

$$\beta = \frac{\pi}{2} - \alpha.$$

(Note that in (10) the notations $x_2 - x_0 = 2\pi + x_2 - x_N$ and $x_{N+1} - x_{N-1} = 2\pi + x_1 - x_{N-1}$ are used to represent periodic boundary conditions).

In other words, whatever stationary coherence-incoherence pattern we find in model (5) with $N \to \infty$, relations (7)–(9) will always be satisfied for it, regardless of the natural frequency $\omega$, the phase lag $\alpha$ and the coupling function $G(x)$. Note that by stationarity we mean here that the pattern does not change its position on the network and does not change its shape at the macroscopic level. Therefore, dynamic patterns such as travelling chimera states[38] and breathing chimera states[39] are currently excluded from our consideration. Moreover, the requirement of pattern stationarity automatically imposes a reflection-symmetry requirement on $G(x)$, since asymmetric coupling functions usually lead to pattern motion[38].

In the following, we want to verify whether relations (7)–(9) can be used to recover the main system parameters based on the observation of a stationary chimera state in model (5). More precisely, we assume that the observation is limited to recording a relatively small dataset, including oscillator positions $x_j$, effective frequencies $\Omega_j$, and local order parameters $\zeta_j$.

## Practical accuracy of SERs

Thermodynamic limit theory predicts that SERs (7)–(9) are only exact for an infinitely large system size $N$ and for effective frequencies $\Omega_j$ and local order parameters $\zeta_j$ calculated by infinitely long time averaging. But they also remain approximately accurate under much weaker constraints. For example, let us consider finite-time averages

$$\Omega_j(T) = \frac{1}{T}\int_0^T \frac{d\theta_j(t)}{dt}dt, \tag{11}$$

$$\zeta_j(T) = \frac{1}{T}\int_0^T e^{i\theta_j(t)}\frac{\overline{Z}(t)}{|Z(t)|}dt, \tag{12}$$

$$\Omega(T) = \frac{1}{T}\int_0^T \mathrm{Im}\left(\frac{1}{Z(t)}\frac{dZ(t)}{dt}\right)dt, \tag{13}$$

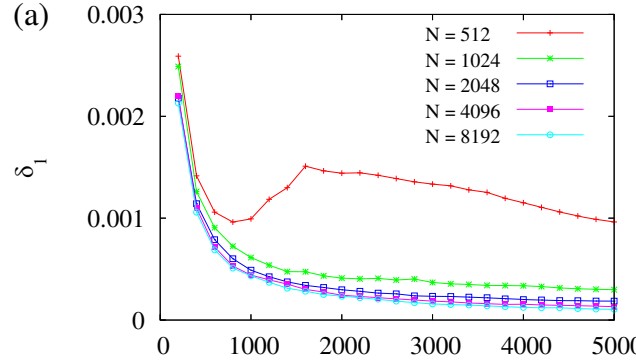

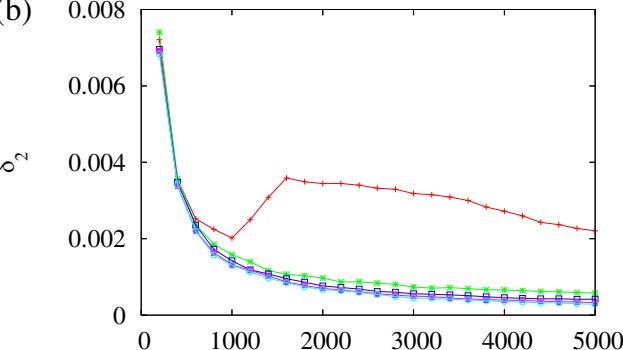

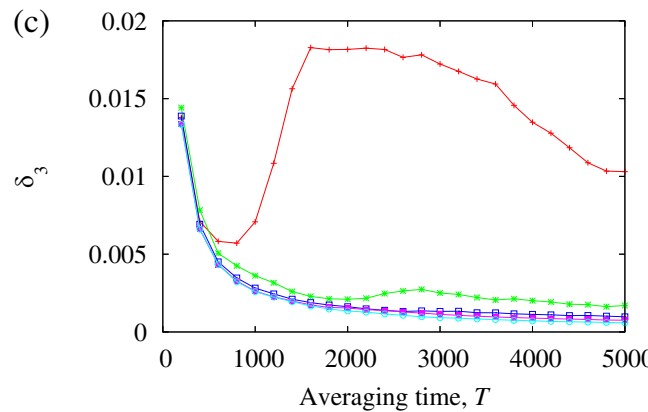

**Fig. 3 | Accuracy of statistical equilibrium relations for finite $N$.** Mean discrepancies of SERs (7)–(9) for the chimera state from Fig. 2. The discrepancies $\delta_1$, $\delta_2$ and $\delta_3$ defined in the text are shown separately in (**a**–**c**) respectively. The five curves in each panel show the results for different system sizes $N$ mentioned in panel (**a**).

and $\xi_j(T)$ given by formula (10) with $\zeta_j(T)$ and $\Omega(T)$. Then, for each of the SERs (7)–(9), we can define its mean discrepancy

$$\delta_1(T) = \frac{1}{N}\sum_{j=1}^{N}\left|\frac{\omega - \Omega_j(T)}{\omega - \Omega(T)} - \frac{2|\zeta_j(T)|^2}{1+|\zeta_j(T)|^2}\right|,$$

$$\delta_2(T) = \frac{1}{N}\sum_{j=1}^{N}\left|\mathrm{Re}\left(\frac{\xi_j(T)}{\zeta_j(T)}\right) - \frac{2}{1+|\zeta_j(T)|^2}\right|,$$

$$\delta_3(T) = \frac{1}{N_*}\sum_{j:|\zeta_j(T)|<1-1/\sqrt{N}}\left|\xi_j(T) - \frac{2\zeta_j(T)}{1+|\zeta_j(T)|^2}\right|,$$

where $N_*$ is the number of indices $j$ satisfying the inequality $|\zeta_j(T)| < 1/\sqrt{N}$. Calculating these mean discrepancies for the chimera state from Fig. 2, as well as for chimera states with the same parameters but different system sizes $N$, we see that SERs (7)–(9) are very accurate already for $N > 1000$ and averaging times $T > 1000$, Fig. 3.

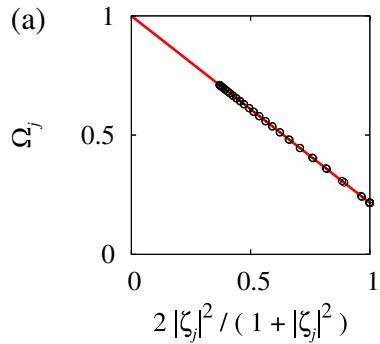

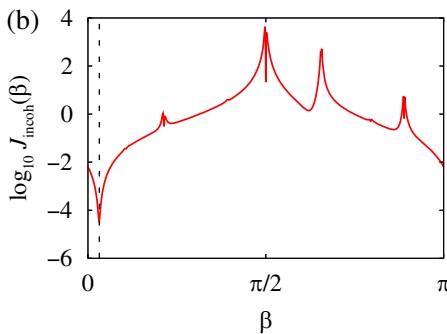

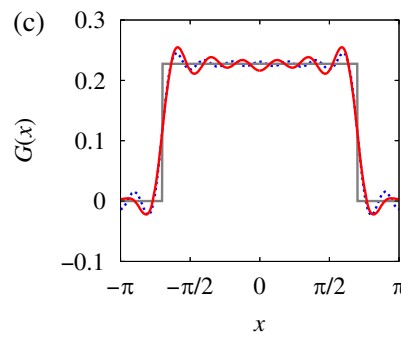

**Fig. 4 | Reconstruction of model parameters using statistical equilibrium relations (SER). a** SER (7) for the chimera state in Fig. 2. The circles show the averages calculated from the numerical trajectory (only every 16th point is shown), the line shows a linear fit. **b** The graph of the function $J_{incoh}(\beta)$. The dashed line shows the position of the minimum $\beta_{min}$. **c** The red/dark curve shows the reconstructed coupling function with $M = 10$ spatial Fourier modes. The grey/light curve shows the original coupling function $G(x)$ and the dotted curve shows its exact Fourier approximation with 10 modes.

Thus, these relations can also be used in realistic situations where $N$ and $T$ are moderately large. This approach is roughly comparable to the application of the laws of thermodynamics, which are proven by statistical physics for infinitely large systems, but are used for systems consisting of a finite number of particles, provided that this number is large enough.

**Parameter reconstruction algorithm**

Suppose that the statistical equilibrium relations (7)–(9) are satisfied (with some discrepancy) for the observables $\Omega_j$ and $\zeta_j$. How can we use this fact to reconstruct the parameters $\omega$, $\beta$ and $G(x)$ in model (5)? It is easy to see that the values of $\omega$ and $\omega - \Omega$ can be found from the statistical equilibrium relations (7) written in the form

$$\Omega_j = \omega + (\Omega - \omega)\eta_j \text{ with } \eta_j = 2|\zeta_j|^2/(1+|\zeta_j|^2).$$

Indeed, standard linear regression yields, see Fig. 4a,

$$\omega = \frac{S_\Omega S_{\eta\eta} - S_\eta S_{\Omega\eta}}{S_{\eta\eta} - S_\eta^2}, \quad \Omega - \omega = \frac{S_{\Omega\eta} - S_\eta S_\Omega}{S_{\eta\eta} - S_\eta^2},$$

where

$$S_\eta = \frac{1}{N}\sum_{j=1}^N \eta_j, \quad S_\Omega = \frac{1}{N}\sum_{j=1}^N \Omega_j,$$

$$S_{\eta\eta} = \frac{1}{N}\sum_{j=1}^N \eta_j^2, \quad S_{\Omega\eta} = \frac{1}{N}\sum_{j=1}^N \Omega_j\eta_j.$$

The remaining phase lag parameter $\beta$ and the coupling function $G(x)$ can be found as follows. First, we note that formula (10) implies

$$\text{Re}\left(\frac{\xi_j}{\zeta_j}\right) = \sum_{k=1}^N \frac{G(x_j - x_k)}{\omega - \Omega}\text{Re}\left(e^{i\beta}\frac{\zeta_k}{\zeta_j}\right)\frac{x_{k+1} - x_{k-1}}{2}$$

for all $j = 1, ..., N$. On the other hand, if $G(x)$ is symmetric, i.e. $G(-x) = G(x)$, then it can be approximated by a Fourier sum

$$G(x) = \sum_{m=0}^M c_m q_m(x) \text{ with } q_m(x) = \cos(mx). \quad (14)$$

In Supplementary Note 1, we explain that for each chimera state there is an optimal number of spatial modes $M_{opt}$ that corresponds to the best efficiency of the reconstruction algorithm. But it is not known a priori, so $M$ in Eq. (14) is chosen empirically.

According to (8) we can expect that the vector $(\{c_m\}, \beta)$ is the minimizer of the functional

$$J(\{c_m\}, \beta) = \frac{1}{N}\sum_{j=1}^N \left[\frac{2}{1+|\zeta_j|^2} - \sum_{m=0}^M c_m Q_{jm}(\beta)\right]^2,$$

where

$$Q_{jm}(\beta) = \sum_{k=1}^N \frac{q_m(x_j - x_k)}{\omega - \Omega}\text{Re}\left(e^{i\beta}\frac{\zeta_k}{\zeta_j}\right)\frac{x_{k+1} - x_{k-1}}{2}.$$

Note that in order not to lose the information provided by relations (9), we use the minimization problem for $J(\{c_m\}, \beta)$ only to express the coefficients $c_m$ as functions of $\beta$. For this, we rewrite the corresponding local minimum condition

$$\partial_{c_n} J(\{c_m\}, \beta) = -\frac{2}{N}\sum_{j=1}^N Q_{jn}(\beta)\left[\frac{2}{1+|\zeta_j|^2} - \sum_{m=0}^M c_m Q_{jm}(\beta)\right] = 0$$

in the matrix form

$$A(\beta)c = b(\beta), \quad (15)$$

where

$$A_{nm}(\beta) = \sum_{j=1}^N Q_{jn}(\beta)Q_{jm}(\beta)$$

and

$$b_n(\beta) = \sum_{j=1}^N \frac{2Q_{jn}(\beta)}{1+|\zeta_j|^2}.$$

Then, the solution of (15) reads

$$\widetilde{c}(\beta) = A^{-1}(\beta)b(\beta).$$

Now, using the statistical equilibrium relations (9) and formulas (10) and (14), we define a function

$$J_{incoh}(\beta) = \frac{1}{N}\sum_{j:|\zeta_j|<1-\frac{1}{\sqrt{N}}} \left|\frac{2\zeta_j}{1+|\zeta_j|^2} - \sum_{m=0}^M e^{i\beta}\widetilde{c}_m(\beta)\widetilde{Q}_{jm}\right|^2,$$

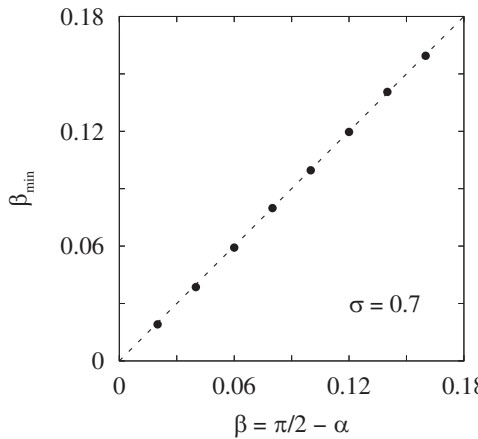

**Fig. 5 | Efficiency of the proposed method in reconstructing the phase lag β.** The method was applied to the chimera state observed in model (5) with top-hat coupling. The dots show the values of $\beta_{\min}$ found as the global minimum of the function $J_{\text{incoh}}(\beta)$ for different β. Parameters: $N = 2048$, $\omega = 1$ and $\sigma = 0.7$.

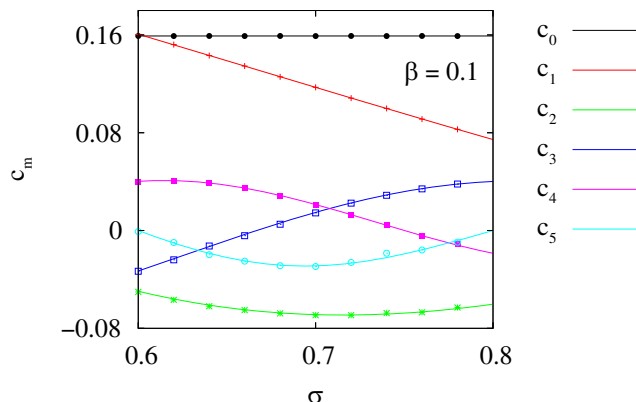

**Fig. 6 | Efficiency of the proposed method in reconstructing the coupling function G(x).** The six leading Fourier coefficients $c_m$ in formula (14) were recovered from the chimera state observed in model (5) with top-hat coupling. The curves show the theoretical values given by formula (16) and the symbols show the reconstructed values. Parameters: $N = 2048$, $\omega = 1$ and $\beta = 0.1$.

where

$$\widetilde{Q}_{jm} = \sum_{k=1}^{N} \frac{q_m(x_j - x_k)}{\omega - \Omega} \zeta_k \frac{x_{k+1} - x_{k-1}}{2},$$

and look for its global minimum $\beta_{\min}$, which in theory must coincide with the value of phase lag β in model (5). (Note that in the thermodynamic limit, statistical equilibrium relation (9) holds for all oscillators $j$ with $|\zeta_j| < 1$. But because of the finite-size fluctuations we replace this inequality with a more restrictive one $|\zeta_j| < 1 - 1/\sqrt{N}$ in our definition of $J_{\text{incoh}}(\beta)$.) To find the global minimum of $J_{\text{incoh}}(\beta)$, we calculate this function at 20 points evenly spaced in the interval $[0, \pi]$ and use the resulting approximate estimate of the minimizer as an initial guess to solve the equation $J'_{\text{incoh}}(\beta) = 0$ using Newton's method. The obtained global minimizer $\beta_{\min}$ is interpreted as an approximate value of the phase lag β in Eq. (5), see Fig. 4b. Respectively, formula (14) with $c_m = \tilde{c}_m(\beta_{\min})$ gives an approximate representation of the coupling function $G(x)$, see Fig. 4c.

To complete the description of our algorithm, we will also estimate its computational cost. It is easy to see that the largest matrices used by the algorithm are the $N \times M$ matrices $Q_{jm}(\beta)$ and $\widetilde{Q}_{jm}(\beta)$. All their elements can be computed in $O(N^2M)$ operations, while the entire matrix $A(\beta)$ in Eq. (15) can be computed in $O(NM^2)$ operations. Moreover, solving the linear system (15) requires an additional $O(M^3)$ operations. Together, this means that the minimization of the functionals $J(\{c_m\}, \beta)$ and $J_{\text{incoh}}(\beta)$ requires $O(N^2M + NM^2 + M^3)$ operations. Therefore, if $M \ll N$, the resulting computational cost is much less than the standard number of operations $O(N^3)$ required to invert an $N \times N$ matrix.

## Examples

To illustrate the possibilities of our parameter reconstruction algorithm, we apply it to the analysis of chimera states in model (5) with top-hat coupling and $N = 2048$. We choose the natural frequencies of all oscillators to be $\omega = 1$. Then, for a fixed coupling range $\sigma = 0.7$, we vary the phase lag β in the range from 0.02 to 0.16, where stable chimera states can be observed. For each value of β, we first simulate system (5) for $10^5$ time units to allow it to reach statistical equilibrium, and then calculate the effective frequencies $\Omega_j$ and local order parameters $\zeta_j$ using formulas (11) and (12) with $T = 2000$. Finally, we apply the above described parameter reconstruction algorithm with $M = 10$ spatial Fourier harmonics in formula (14). Figure 5 shows that in this way the value of β is reconstructed with an absolute accuracy of less

than 0.0015. Similarly, the value of ω is reconstructed with an accuracy of less than 0.0008 (not shown).

In another round of simulations, we fix $\beta = 0.1$ and vary the coupling range σ from 0.6 to 0.78. For each value of σ, we repeat the same numerical protocol as above and check the accuracy with which our algorithm reconstructs the six leading Fourier coefficients $c_m$ in formula (14). Note that for the top-hat coupling function, these coefficients can be calculated analytically

$$c_m = \begin{cases} 1/(2\pi) & \text{for } m = 0, \\ \sin(\pi m \sigma)/(\pi^2 m \sigma) & \text{for } m = 1, 2, \ldots, \end{cases} \tag{16}$$

so in Fig. 6 we compare the theoretical curves with several reconstructed parameter values (symbols), which turn out to be in excellent agreement with each other.

Finally, in Figs. 7–9 we show that the proposed parameter reconstruction algorithm works equally well for other types of coupling functions in model (5). In particular, comparing the distribution of Fourier coefficients $c_m$ in Fig. 6, Fig. 8 and Fig. 9, we clearly see the possibility of distinguishing nonlocal couplings of the top-hat, exponential and cosine type. Moreover, from the value of the Fourier coefficient $c_1$, we can uniquely determine the ranges and decay rates of the corresponding coupling functions, which confirms the reliability and efficiency of the proposed approach.

It is important to note that although all the coupling functions $G(x)$ considered above are normalized to unity, this is not a requirement of our method. We simply followed the established tradition in the literature on chimera states. Coupling functions with other normalizations can be analysed in the same way.

## Parameter reconstruction algorithm with partial data

Even though the reconstruction algorithm described above requires performing calculations with only $2N$ variables $\Omega_j$ and $\zeta_j$, it can still become too resource-demanding if $N$ is too large. This complication can be overcome by noting that statistical equilibrium relations (7)–(9) also remain valid if, instead of all values $(x_j, \Omega_j, \zeta_j)$, $j = 1, \ldots, N$, only a sufficiently large subset of them is used. Roughly speaking, from $N$ points $x_j$ we can randomly select a smaller subset $\{x_j : j \in S\}$ with the number of elements $\#\{S\} < N$. Then using the trapezoidal rule we can write an analogue of formula (10) that approximates the integral (21), albeit with worse accuracy than (10) (see Methods). This fact allows us to repeat all the steps of the above reconstruction algorithm, using only the

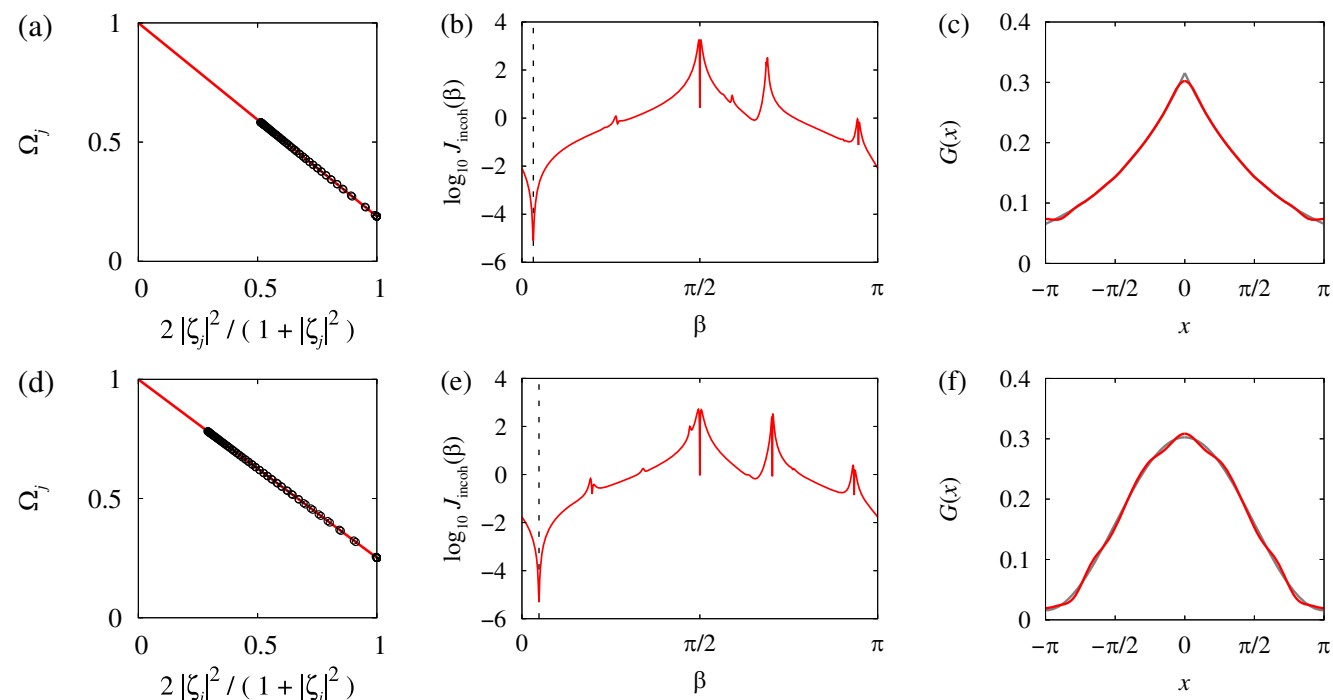

**Fig. 7 | Other examples of application of the parameter reconstruction algorithm to chimera states in model (5) with** $N = 2048$. **a–c** Exponential coupling function with $\kappa = 0.5$, $\omega = 1$, and $\alpha = \pi/2 - 0.1$. **d–f** Cosine coupling function with $A = 0.9$, $\omega = 1$, and $\alpha = \pi/2 - 0.15$. In both cases, the coupling function $G(x)$ was approximated by the ansatz (14) containing $M = 10$ spatial Fourier modes. In (**c**, **f**), the red/dark curves show reconstructed coupling functions, while the grey/light curves show the original coupling functions. Other notations are the same as in Fig. 4.

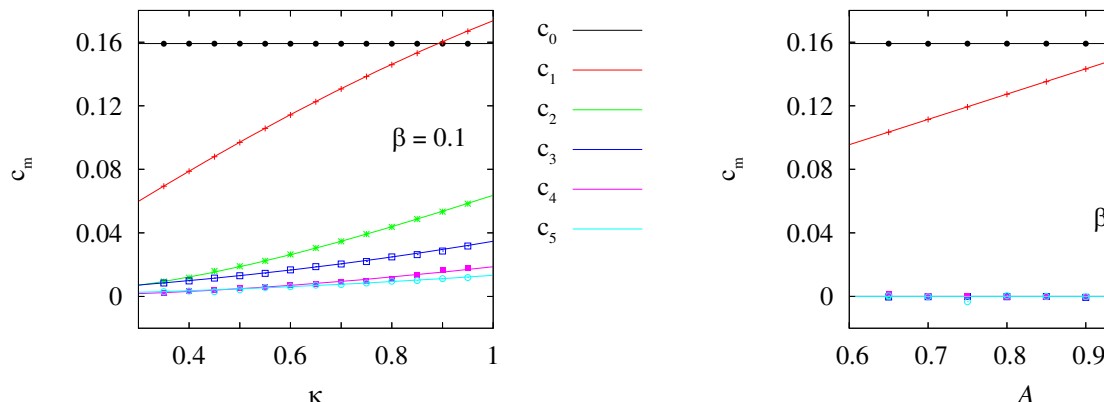

**Fig. 8 | Reconstruction of the exponential coupling function from the chimera state observed in model (5).** The Fourier coefficients, given by the formulas $c_0 = 1/(2\pi)$ and $c_m = (1 - (-1)^m e^{-\pi\kappa})\kappa^2/(\pi(1 - e^{-\pi\kappa})(\kappa^2 + m^2))$ for $m \geq 1$, are shown as curves, and the symbols indicate the reconstructed values. Other parameters: $N = 2048$, $\omega = 1$ and $\beta = 0.1$.

**Fig. 9 | Reconstruction of the cosine coupling function from the chimera state observed in model (5).** The Fourier coefficients, given by the formulas $c_0 = 1/(2\pi)$, $c_1 = A/(2\pi)$ and $c_m = 0$ for $m \geq 2$, are shown as curves, and the symbols indicate the reconstructed values. Other parameters: $N = 2048$, $\omega = 1$ and $\beta = 0.15$.

indices $j \in S$ in the linear regression, as well as in the definition of $J(\{c_m\}, \beta)$ and $J_{\text{incoh}}(\beta)$. Importantly, in this case, we need to calculate the effective frequencies and local order parameters only for $j \in S$. Moreover, the global order parameter $Z(t)$ in (12) must be replaced with its "rarefied" analogue

$$Z(t) = \frac{1}{\#\{S\}} \sum_{j \in S} e^{i\theta_j(t)}.$$

Thus, the resulting reconstruction algorithm will use only observation of oscillators $\theta_j(t)$ with $j \in S$.

Figure 10 shows how such a modified algorithm works for a chimera state in model (5) with top-hat coupling and $N = 8192$ oscillators, if instead of all oscillators we randomly select 25% of them. Comparing Figs. 4 and 10, we see that our algorithm has good performance also with partial data.

### Time sampling and sensitivity to measurement noise

The only input data used in our algorithm are the time-averaged values of $\Omega_j$ and $\zeta_j$, which can be considered its advantage. Indeed, time averaging is a natural low-pass filter, so the algorithm is insensitive to the presence of noise in the phases $\theta_j$, provided that the noise is unbiased (i.e., has a zero mean). On the other hand, for time averaging,

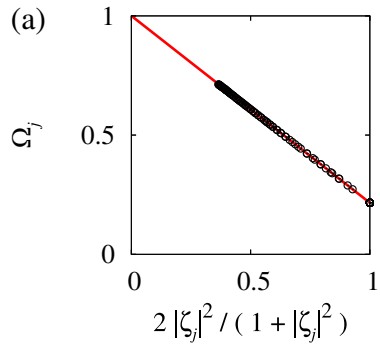
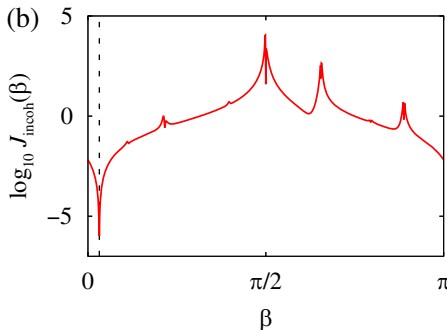
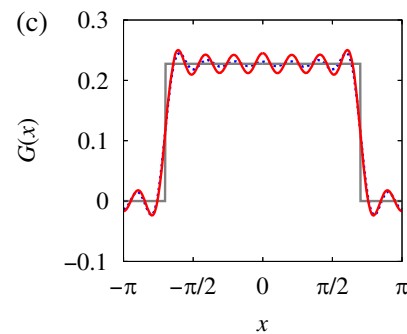

**Fig. 10 | Application of the parameter reconstruction algorithm to partial data.** For a chimera state in model (5) with top-hat coupling and $N = 8192$ oscillators, we used 1996 randomly selected oscillators to reconstruct the parameters $\omega$, $\beta$ and $G(x)$. The quantities shown in (**a**–**c**) are the same as in Fig. 4. Other model parameters are given in Fig. 2.

phases do not need to be measured at evenly spaced time points (as is done in the map-based algorithms in refs. 40,41). More precisely, if we calculate the local order parameter $\zeta_j$ by formula

$$\zeta_j = \frac{1}{N_T} \sum_{k=1}^{N_T} e^{i\theta_j(t_k)} \frac{\overline{Z}(t_k)}{|Z(t_k)|},$$

then we only need to worry that the number of points $N_T$ is large enough and that the points $t_k$ are uniformly distributed with respect to the oscillation period. As for calculating the effective frequencies $\Omega_j$, the effective formula is

$$\Omega_j = \frac{1}{t_{N_T} - t_1} \sum_{k=2}^{N_T} \arg e^{i(\theta_j(t_k) - \theta_j(t_{k-1}))}.$$

Here, we need to satisfy the Nyquist criterion that the minimum of the intervals $t_k - t_{k-1}$ is less than the half-period of the corresponding oscillations. But the intervals $t_k - t_{k-1}$ do not have to be small, since we are not computing any time derivatives.

### Chimera states with zero global order parameter

The definition of local order parameters exploits the fact that the global order parameter $Z(t)$ does not vanish, and therefore its argument $Z(t)/|Z(t)|$ is well-defined. However, this condition is not satisfied for all types of chimera states. For example, it is violated for so called multi-headed chimera states[42] with antiphase adjacent coherent regions. Fortunately, in this case, an alternative definition of local order parameters can be given. To do this, we first note that for patterns with antiphase regions, the Daido order parameter

$$Z_2(t) = \frac{1}{N} \sum_{k=1}^{N} e^{2i\theta_k(t)}$$

is usually non-vanishing. But the argument of this order parameter has twice the speed compared to the order parameter $Z(t)$. Therefore, we define

$$\widetilde{\zeta}_j = \left\langle e^{i\theta_j(t)} \sqrt{\frac{\overline{Z_2(t)}}{|Z_2(t)|}} \right\rangle,$$

where the complex square root is calculated in such a way that it varies as a continuous function of time. The value of $\widetilde{\zeta}_j$ may or may not coincide with the value of $\zeta_j$ defined in the Introduction, depending on the choice of the square root branch at $t = 0$. To avoid

this ambiguity, we ultimately define

$$\zeta_j = \widetilde{\zeta}_j \frac{\overline{\widetilde{Z}}}{|\widetilde{Z}|}, \quad \text{where } \widetilde{Z} = \frac{1}{N} \sum_{k=1}^{N} \widetilde{\zeta}_k.$$

The resulting formula gives the correct values of the local order parameters for multi-headed chimera states, using which the parameter reconstruction algorithm can be applied to these coherence-incoherence patterns as well.

### Calculations with protophases

In applications, phase oscillator networks are used as simplified mathematical models to study the behaviour of weakly interacting limit cycle oscillators. The physical phase of a limit cycle oscillator is defined as a variable that increases uniformly from 0 to $2\pi$ during one cycle of its oscillations[10,11]. It is this phase that appears in the formulation of Kuramoto-type models obtained using the phase reduction method[10,13–15]. But the physical phase cannot be measured directly in an experiment. Therefore, it is usually extracted from the measured signal using Hilbert transform, projection onto a two-dimensional plane, marker appearance analysis, etc. In each of these approaches, the protophase is obtained, i.e., a quantity of the form $\phi = \Phi(\theta)$ where $\theta$ is the physical phase and $\Phi(\theta)$ is an unknown $2\pi$-periodic phase-protophase transformation[43]. This means that additional effort is usually required to reconstruct the function $\Phi(\theta)$ along with other system parameters. Fortunately, for stationary coherence-incoherence patterns, such a step is not needed, provided that we only want to find the effective frequencies and local order parameters. This can be done as follows.

Suppose that we found a chimera state in Eq. (5), and instead of the physical phases $\theta_j(t)$ we have as available data the corresponding protophases $\phi_j(t) = \Phi(\theta_j(t))$. Then, thanks to (32), the effective frequencies $\Omega_j$ can be approximately calculated by

$$\Omega_j = \frac{1}{T} \int_0^T \frac{d\phi_j(t)}{dt} dt.$$

Recalling that the coherent region of the chimera state corresponds to a plateau in the graph of $\Omega_j$ versus $x_j$, see Fig. 2, we can easily identify its frequency, for example, as a minimum $\Omega_c = \min_j \Omega_j$. Moreover, according to the continuum limit analysis presented in Methods, the frequency $\Omega_c$ coincides with the frequency of the global order parameter $Z(t)$, so that $Z(t) \approx Z_0 e^{i\Omega_c t}$.

Next, we consider the modified local order parameters

$$\widehat{\zeta}_j = \frac{1}{T} \int_0^T e^{i\phi_j(t)} e^{-i\Omega_c t} dt.$$

As explained in Methods, there are simple formulas (33), (34), independent of the explicit form of $\Phi(\theta)$, which allow us to express the local order parameters $\zeta_j$ from their modified counterparts $\widehat{\zeta}_j$, provided that $N \gg 1$ and the averaging time $T$ is sufficiently long. Using these formulas as an approximation in the case of moderate size $N$ and moderate averaging time $T$, we can write

$$\zeta_j = \frac{\widehat{\zeta}_j}{\max_k |\widehat{\zeta}_k|} \frac{\overline{\widehat{Z}}}{|\widehat{Z}|}, \quad \text{where } \widehat{Z} = \frac{1}{N} \sum_{j=1}^{N} \widehat{\zeta}_j.$$

Thus, all the input information needed for our parameter reconstruction algorithm can ultimately be obtained from the protophases $\phi_j(t)$ without knowing the explicit form of the phase-protophase transformation $\Phi(\theta)$.

## Discussion

The problem of model reconstruction for dynamical systems capable of exhibiting various patterns of synchrony and disorder has been a subject of research for a long time. Two general questions have usually been in focus. What is the coupling topology (i.e., network architecture) between individual agents in the system?[44] And what form of coupling functions describes the interaction between these agents?[45] Various methods have been proposed to answer these questions, including the finite-time mapping approach[40,41], and random phase resetting method[46], fixed points analysis[44], and kernel density estimation[47]. In[48] it was shown that for pulse-coupled oscillators the network topology can be reconstructed from spiking sequences. In addition, some methods were inspired by data assimilation approach, including dynamical Bayesian inference[49], maximum likelihood estimation combined with multiple shooting[50,51], and an ensemble Kalman filter with state space augmentation[52]. Each of these methods has its advantages and disadvantages, but they all become increasingly complex and resource-demanding as the system size increases, so they are usually applied to systems consisting of several dozen or hundreds of individual agents. On the contrary, in this paper we described a method that is much better suited for similar inverse problems, but in the case of large-size systems. We showed how it can be used to noninvasively reconstruct the parameters of the Kuramoto-Battogtokh model from a single observation of a chimera state transformed into a small dataset of time-averaged quantities. Although we have only examined this special example in detail, it seems that the method can also be generalized to a broader class of networks, including two-dimensional arrays with nonlocal coupling[53,54], as well as networks with heterogeneous coupling coefficients[55] and heterogeneous natural frequencies[56]. (Several examples of how this can be done are described in the section Methods below). Moreover, given the existing continuum limit theory for travelling and breathing chimera states[38,39], the SER-based approach can also be extended to analyze these nonstationary coherence-incoherence patterns, which remained outside the scope of the present work. Furthermore, using our method, it is potentially possible to consider phase oscillator models with higher-harmonics[57] and higher-order interactions[58,59], although in this case other types of time-averaged observables and multiple observations would certainly be required.

From a more general perspective, analogues of statistical equilibrium relations can be written not only for phase oscillator networks, but also for many other systems, in particular for those that can be considered using the Ott-Antonsen theory or the self-consistency approach (see Methods). This suggests that the proposed model reconstruction scheme can be applied with appropriate modifications to neural networks (e.g., those consisting of theta neurons[60,61] or quadratic integrate-and-fire neurons[62,63]) and Kuramoto-type models for power grids[64,65].

Finally, we note that the knowledge of the existence of statistical equilibrium relations in a given system can be useful in itself. For example, it can be a natural clue to the lower-dimensional manifold or collective variables representing its long-term dynamics[4,5,66]. Such information, in turn, can facilitate or refine the development of a data-driven model reconstruction algorithms, reducing their memory usage and increasing their computational efficiency.

## Methods
### Derivation of SERs for nonlocally coupled oscillators

In this section, we will show how to derive statistical equilibrium relations (7)–(9) for the Kuramoto-Battogtokh system (5). Importantly, we do not make any special assumptions about the natural frequency $\omega$, the phase lag $\alpha$, or the coupling function $G(x)$. But we assume that some stationary coherence-incoherence pattern arises in system (5). Roughly speaking, we carry out the following steps. First, we write the continuity equation corresponding to system (5) in the large-$N$ limit and the general self-consistent ansatz of its stationary solutions. Using this ansatz, we obtain formulas (29), (30) and (31), which have the form of statistical equilibrium relations. Finally, to complete the definition of relations (29) and (30), we write down the Riemann sum approximation of formula (21), which gives the analytic expression (26) for the mean field $\xi_j$. Note that the above derivation scheme can be easily generalized to other types of coupled oscillator systems, for which the continuity equation and a self-consistent representation of its stationary solutions can be written.

First, we rewrite the Kuramoto-Battogtokh system (5) in the form (1), (2):

$$\frac{d\theta_j}{dt} = \omega - \text{Im}\left(\overline{W}_j e^{i\theta_j} e^{i\alpha}\right), \tag{17}$$

where

$$W_j = \frac{2\pi}{N} \sum_{k=1}^{N} G(x_j - x_k) e^{i\theta_k} \tag{18}$$

and $\overline{W}_j$ denotes the complex-conjugate of $W_j$.

In the large-$N$ limit, called also the continuum limit, the state of the system (5) can be represented by a probability density function $\rho(\theta, x, t)$ with $x \in [-\pi, \pi]$. Then, the dynamics of $\rho$ is determined by a continuity equation

$$\frac{\partial \rho}{\partial t} + \frac{\partial}{\partial \theta} \left( \left[ \omega - \int_{-\pi}^{\pi} \int_{-\pi}^{\pi} G(x - x') \sin(\theta - \theta' + \alpha) \right. \right.$$
$$\left. \left. \times \rho(\theta', x', t) d\theta' dx' \right] \rho \right) = 0.$$

It is known[27,67] that the chimera patterns shown above behave like statistical equilibria, namely, each of them has a time-independent probability density in an appropriate corotating frame. In addition, it is known that these probability densities lie on a special Ott-Antonsen manifold[68,69] consisting of functions of the form

$$\rho(\theta, x, t) = \frac{1}{2\pi} \left( 1 + \sum_{n=1}^{\infty} \left[ \overline{z}^n(x, t) e^{in\theta} + z^n(x, t) e^{-in\theta} \right] \right),$$

where $z(x, t)$ satisfies the integro-differential equation

$$\frac{dz}{dt} = i\omega z + \frac{1}{2} e^{-i\alpha} \mathcal{G}z - \frac{1}{2} e^{i\alpha} z^2 \mathcal{G}\overline{z} \tag{19}$$

with the integral operator

$$(\mathcal{G}z)(x,t) = \int_{-\pi}^{\pi} G(x-x')z(x',t)dx',$$

and moreover $|z(x,t)| \leq 1$ for all $x \in [-\pi, \pi]$.

In ref. 67 it was shown that every stationary chimera state in the Kuramoto-Battogtokh system (5) corresponds to a rotating wave solution of Eq. (19) given by the formula

$$z(x,t) = a(x)e^{i\Omega t}. \tag{20}$$

Inserting this ansatz into Eq. (19) and denoting

$$w(x) = \frac{1}{\omega - \Omega} \int_{-\pi}^{\pi} G(x-x')a(x')dx', \tag{21}$$

we obtain

$$e^{-i\beta}\overline{w}(x)a^2(x) - 2a(x) + e^{i\beta}w(x) = 0. \tag{22}$$

These equations allow us to justify the statistical equilibrium relations (7)−(9).

Note that the above ansatz for $\rho(\theta, x, t)$ implies

$$\int_{-\pi}^{\pi}\int_{-\pi}^{\pi} \rho(\theta,x,t)d\theta\,dx = 2\pi,$$
$$\int_{-\pi}^{\pi} e^{ik\theta}\rho(\theta,x,t)d\theta = a^k(x)e^{ik\Omega t} \text{ for } k \in \mathbb{N}. \tag{23}$$

Therefore, in the large-$N$ limit, the definition of the global order parameter (6) can be written in the form

$$Z(t) = \frac{1}{2\pi}\int_{-\pi}^{\pi}\int_{-\pi}^{\pi} e^{i\theta}\rho(\theta,x,t)d\theta\,dx = Z_0 e^{i\Omega t},$$

where

$$Z_0 = \frac{1}{2\pi}\int_{-\pi}^{\pi} a(x)dx.$$

Similarly, using the ergodicity property, we obtain

$$\zeta_j = \int_{-\pi}^{\pi} e^{i\theta}\frac{\overline{Z}_0}{|Z_0|}e^{-i\Omega t}\rho(\theta,x_j,t)d\theta = a(x_j)\frac{\overline{Z}_0}{|Z_0|}, \tag{24}$$

and therefore

$$\frac{1}{N}\sum_{j=1}^{N}\zeta_j = |Z_0| \text{ as } N \to \infty. \tag{25}$$

Let us denote $a_j = a(x_j)$ and $w_j = w(x_j)$, then using the trapezoidal rule we write an approximate version of the definition (21)

$$w_j = \frac{1}{\omega - \Omega}\sum_{k=1}^{N} G(x_j - x_k)a_k \frac{x_{k+1} - x_{k-1}}{2}, \tag{26}$$

where due to the periodicity of the variable $x$ we assume $x_2 - x_0 = 2\pi + x_2 - x_N$ and $x_{N+1} - x_{N-1} = 2\pi + x_1 - x_{N-1}$. Multiplying this by $e^{i\beta}\overline{Z}_0/|Z_0|$ and defining

$$\xi_j = w_j e^{i\beta}\frac{\overline{Z}_0}{|Z_0|}, \tag{27}$$

we obtain formula (10). On the other hand, from Eq. (22) it follows

$$\overline{\xi}_j\zeta_j^2 - 2\zeta_j + \xi_j = 0. \tag{28}$$

**Proposition.** Suppose that $\xi_j \in \mathbb{C}$ and $\zeta_j \in \mathbb{C}$ satisfy equation (28), then

$$\xi_j = \frac{2\zeta_j}{1 + |\zeta_j|^2} \text{ for } |\zeta_j| \neq 1 \tag{29}$$

and

$$\mathrm{Re}(\xi_j\overline{\zeta}_j) = 1 \text{ for } |\zeta_j| = 1. \tag{30}$$

Moreover, for all values of $|\zeta_j|$ we have

$$\mathrm{Re}\left(\frac{\xi_j}{\zeta_j}\right) = \frac{2}{1 + |\zeta_j|^2}$$

and

$$\mathrm{Re}\left(\xi_j\overline{\zeta}_j\right) = \frac{2|\zeta_j|^2}{1 + |\zeta_j|^2}.$$

**Proof:** The complex conjugate of Eq. (28) reads

$$\xi_j\overline{\zeta}_j^2 - 2\overline{\zeta}_j + \overline{\xi}_j = 0,$$

or equivalently $\overline{\xi}_j = 2\overline{\zeta}_j - \xi_j\overline{\zeta}_j^2$. Inserting this into Eq. (28), we obtain

$$2|\zeta_j|^2\zeta_j - \xi_j|\zeta_j|^4 - 2\zeta_j + \xi_j = 0,$$

or

$$\xi_j(1 - |\zeta_j|^4) = 2\zeta_j(1 - |\zeta_j|^2).$$

If $|\zeta_j| \neq 1$, this yields (29).

On the other hand, if $|\zeta_j| = 1$, then $\overline{\zeta}_j = 1/\zeta_j$, and therefore dividing (28) by $\zeta_j$, we obtain

$$\overline{\xi}_j\zeta_j - 2 + \xi_j\overline{\zeta}_j = 0$$

what is equivalent to (30). ■

When $N \gg 1$, formula (18) can be rewritten using (23), (26) and (27). This gives

$$W_j = \frac{2\pi}{N}\sum_{k=1}^{N} G(x_j - x_k)a(x_k)e^{i\Omega t} = (\omega - \Omega)w_j e^{i\Omega t}$$
$$= (\omega - \Omega)\xi_j e^{-i\beta}\frac{Z_0}{|Z_0|}e^{i\Omega t}.$$

Inserting this into (17), we obtain

$$\frac{d\theta_j}{dt} = \omega - (\omega - \Omega)\mathrm{Im}\left(i\overline{\xi}_j\frac{\overline{Z}_0}{|Z_0|}e^{-i\Omega t}e^{i\theta_j}\right).$$

Therefore, thanks to the ergodicity property, identity (24) and the above Proposition, we have

$$\Omega_j = \left\langle\frac{d\theta_j}{dt}\right\rangle = \omega - (\omega - \Omega)\mathrm{Im}\left(i\overline{\xi}_j\zeta_j\right)$$
$$= \omega - (\omega - \Omega)\mathrm{Re}\left(\xi_j\overline{\zeta}_j\right) = \omega - (\omega - \Omega)\frac{2|\zeta_j|^2}{1 + |\zeta_j|^2}, \tag{31}$$

that is equivalent to the statistical equilibrium relation (7).

## Formulas involving protophases

In addition to the identities we have already obtained, we can also derive similar formulas containing an arbitrary $2\pi$-periodic function $\Phi(\theta)$ that maps the interval $[0, 2\pi)$ onto itself. More precisely, suppose that $\phi_j(t) = \Phi(\theta_j(t))$, then

$$\left\langle \frac{d\phi_j}{dt} \right\rangle = \left\langle \frac{d\theta_j}{dt} \right\rangle = \Omega_j. \qquad (32)$$

Roughly speaking, both of the above time-averages determine the same rotation number, which is invariant with respect to the form of the function $\Phi(\theta)$.

Next, we consider a modified local order parameter

$$\widehat{\zeta}_j = \left\langle e^{i\phi_j(t)} e^{-i\Omega t} \right\rangle.$$

We note that for any $2\pi$-periodic function $\Phi(\theta)$, it holds

$$e^{i\Phi(\theta)} = \sum_{k=-\infty}^{\infty} \varphi_k e^{ik\theta} \text{ with some } \varphi_k \in \mathbb{C},$$

and therefore

$$\widehat{\zeta}_j = \left\langle e^{i\Phi(\theta_j(t))} e^{-i\Omega t} \right\rangle = \sum_{k=-\infty}^{\infty} \varphi_k \left\langle e^{ik\theta_j(t)} e^{-i\Omega t} \right\rangle.$$

Using the ergodicity property and formulas (23), we calculate

$$\left\langle e^{ik\theta_j(t)} e^{-i\Omega t} \right\rangle = \left\langle \int_{-\pi}^{\pi} e^{ik\theta} e^{-i\Omega t} \rho(\theta, x_j, t) d\theta \right\rangle = \left\langle a^k(x_j) e^{i(k-1)\Omega t} \right\rangle$$

$$= a(x_j)\delta_{k1} = \frac{Z_0}{|Z_0|} \zeta_j \delta_{k1} \text{ for } k \in \mathbb{N},$$

and similarly

$$\left\langle e^{ik\theta_j(t)} e^{-i\Omega t} \right\rangle = 0 \text{ for } k = 0, -1, -2, \ldots.$$

This means

$$\widehat{\zeta}_j = \varphi_1 \frac{Z_0}{|Z_0|} \zeta_j.$$

Defining

$$\widehat{Z} = \frac{1}{N} \sum_{j=1}^{N} \widehat{\zeta}_j \qquad (33)$$

and using (25), we obtain $\widehat{Z} = \varphi_1 Z_0$ and

$$\widehat{\zeta}_j \frac{\overline{\widehat{Z}}}{|\widehat{Z}|} = |\varphi_1| \zeta_j.$$

Given that for each coherent oscillator we have $|\zeta_j| = 1$, and for each incoherent oscillator $|\zeta_j| < 1$, we write $|\varphi_1| = \max_j |\widehat{\zeta}_j|$. Thus, we obtain the formulas

$$\zeta_j = \frac{\widehat{\zeta}_j}{\max_k |\widehat{\zeta}_k|} \frac{\overline{\widehat{Z}}}{|\widehat{Z}|}, \qquad (34)$$

which are valid for every stationary coherence-incoherence pattern in Eq. (5) with $N \gg 1$, regardless of the choice of the function $\Phi(\theta)$.

## Towards other applications

To demonstrate the versatility of the proposed method, we show how it can be applied to other partially synchronized patterns and other coupled oscillator networks. At the same time, we will show that statistical equilibrium relations may look different in different models, and therefore the parameter reconstruction algorithm must be adapted to each specific case individually.

We recall that a plethora of thermodynamic limit results are known for heterogeneous phase oscillator networks, in particular for a network of the form

$$\frac{d\theta_j}{dt} = \omega_j - \frac{1}{N} \sum_{k=1}^{N} q_j q_k \sin(\theta_j - \theta_k + \alpha), \qquad (35)$$

where the natural frequencies $\omega_j$ and coupling strengths $q_j > 0$ are drawn randomly and independently from distributions $H_\omega(\omega)$ and $H_q(q)$, respectively. In the large-$N$ limit, this model allows a universal representation of all stationary partially synchronized states, which is given by the probability density[55,70,71]

$$\rho(\theta, \omega, q, t) = \frac{H_\omega(\omega) H_q(q)}{2\pi}$$
$$\times \left(1 + \sum_{n=1}^{\infty} \left[\overline{a}^n(\omega, q, t) e^{in\theta} + a^n(\omega, q, t) e^{-in\theta}\right]\right),$$

where

$$a(\omega, q, t) = h\left(\frac{\omega - \Omega}{pq}\right) e^{i\Omega t}$$

and

$$h(s) = \begin{cases} s - i\sqrt{1 - s^2} & \text{for } |s| \leq 1, \\ (1 - \sqrt{1 - s^{-2}})s & \text{for } |s| > 1, \end{cases} \qquad (36)$$

and the parameters $p > 0$ and $-\infty < \Omega < \infty$ satisfy the self-consistency equation

$$p = ie^{-i\alpha} \int_{-\infty}^{\infty} d\omega \int_{0}^{\infty} H_\omega(\omega) H_q(q) h\left(\frac{\omega - \Omega}{pq}\right) q \, dq. \qquad (37)$$

It is easy to see that the definitions of the effective frequencies $\Omega_j$ and the local order parameters $\zeta_j$ given in the Introduction also make sense for model (35). Then, rewriting (35) in the form

$$\frac{d\theta_j}{dt} = \omega_j - \text{Im}(\overline{W} e^{i\theta_j} e^{i\alpha}),$$

where

$$W(t) = \frac{1}{N} \sum_{k=1}^{N} q_k e^{i\theta_k(t)},$$

and using the above formula for $\rho(\theta, \omega, q, t)$, two sets of statistical equilibrium relations can be obtained[56]

$$\zeta_k = \frac{\overline{\mathcal{Z}}_0}{|\mathcal{Z}_0|} h\left(\frac{\omega_k - \Omega}{pq_k}\right), \qquad (38)$$

$$\Omega_k = \Omega + pq_k Q\left(\frac{\omega_k - \Omega}{pq_k}\right), \qquad (39)$$

where

$$\mathcal{Z}_0 = \int\limits_{-\infty}^{\infty} d\omega \int\limits_{0}^{\infty} H_\omega(\omega) H_q(q) h\left(\frac{\omega - \Omega}{pq}\right) dq \qquad (40)$$

and

$$Q(s) = \begin{cases} 0 & \text{for } |s| \leq 1, \\ s\sqrt{1 - s^{-2}} & \text{for } |s| > 1. \end{cases}$$

According to their derivation, relations (38) and (39) are exact for $N \to \infty$ and independent of the distributions $H_\omega(\omega)$ and $H_q(q)$. So, from the perspective of the statistical physics, it is logical to assume that they remain valid with a small error also for finite but large $N$.

## SERs for globally coupled oscillators

Our first example is the paradigmatic Kuramoto-Sakaguchi model[72]

$$\frac{d\theta_j}{dt} = \omega_j - \frac{K}{N}\sum_{k=1}^{N} \sin(\theta_j - \theta_k + \alpha), \qquad (41)$$

which describes the dynamics of all-to-all coupled phase oscillators with different frequencies $\omega_j$. In this case $H_q(q) = \delta(q - \sqrt{K})$, and therefore the self-consistency equation (37) and formula (40) can be written in the form

$$\tilde{p} = iKe^{-i\alpha}\mathcal{Z}_0, \qquad (42)$$

where

$$\tilde{p} = p\sqrt{K} \quad \text{and} \quad \mathcal{Z}_0 = \int_{-\infty}^{\infty} H_\omega(\omega) h\left(\frac{\omega - \Omega}{\tilde{p}}\right) d\omega.$$

Accordingly, the statistical equilibrium relations (38) and (39) take the form

$$\zeta_k = ie^{-i\alpha} h\left(\frac{\omega_k - \Omega}{\tilde{p}}\right), \qquad (43)$$

$$\Omega_k = \Omega + \tilde{p} Q\left(\frac{\omega_k - \Omega}{\tilde{p}}\right). \qquad (44)$$

In[56] it was shown that using relations (42)–(44) all parameters in model (41) can be reconstructed from the observed values of $\zeta_k$ and $\Omega_k$ and from the time-averaged magnitude of the global order parameter $Z(t)$. (Note that formula (43) differs from its counterpart in ref. 56 due to the presence of an additional prefactor $-i$ in the definition of $\zeta_k$ used in ref. 56). Roughly speaking, since the function $h(s)$ satisfies the quadratic equation $h^2 - 2sh + 1 = 0$ and relations (43) hold, the parameter $\alpha$ can be found from an explicitly solvable minimization problem

$$\frac{\pi}{2} - \alpha = \underset{\beta \in [0, \pi]}{\operatorname{argmin}} \sum_{k=1}^{N} \left[\operatorname{Im}\left(\frac{\zeta_k^2 e^{-2i\beta} + 1}{2\zeta_k e^{-i\beta}}\right)\right]^2,$$

and moreover

$$\tilde{s}_k = \frac{\omega_k - \Omega}{\tilde{p}} = \operatorname{Re}\left(\frac{\zeta_k^2 e^{2i\alpha} - 1}{2i\zeta_k e^{i\alpha}}\right).$$

Once the ratios $\tilde{s}_k$ are known, the parameters $\Omega$ and $\tilde{p}$ can be determined by linear regression from the set of equations (44). Then, $\omega_k = \Omega + \tilde{p}\tilde{s}_k$. And finally, from (42) we get the coupling strength

$K = \tilde{p}/|\mathcal{Z}_0|$, where $|\mathcal{Z}_0|$ is replaced by the time-averaged magnitude of the global order parameter $Z(t)$ defined by (6).

## SERs for random networks

Partially synchronized patterns can also be found in random networks of identical coupled phase oscillators. In their simplest form, such networks are defined by equations[9]

$$\frac{d\theta_j}{dt} = \omega_0 - \frac{K}{d_{\text{mean}}}\sum_{k=1}^{N} a_{jk} \sin(\theta_j - \theta_k + \alpha), \qquad (45)$$

where $\{a_{jk}\}_{j,k=1}^{N}$ denotes an adjacency matrix such that $a_{jk} = 1$ if there is a link between the $j$th and $k$th oscillators, and $a_{jk} = 0$ otherwise. The number

$$d_j = \sum_{k=1}^{N} a_{jk}$$

is called the degree of the oscillator $j$, and

$$d_{\text{mean}} = \frac{1}{N}\sum_{j=1}^{N} d_j$$

denotes the mean degree. For densely connected random networks, their dynamics can be described using the annealed approximation[9,73]

$$\frac{d\theta_j}{dt} = \omega_0 - \frac{K}{N}\sum_{k=1}^{N} \frac{d_j}{d_{\text{mean}}}\frac{d_k}{d_{\text{mean}}} \sin(\theta_j - \theta_k + \alpha),$$

or equivalently

$$\frac{d\theta_j}{dt} = \omega_0 - \frac{1}{N}\sum_{k=1}^{N} q_j q_k \sin(\theta_j - \theta_k + \alpha), \qquad (46)$$

where

$$q_j = \frac{\sqrt{K} d_j}{d_{\text{mean}}}.$$

Now model (46) looks like a special case of the auxiliary model (35) with $H_\omega(\omega) = \delta(\omega - \omega_0)$. The corresponding statistical equilibrium relations have the form

$$\zeta_k = \frac{\overline{\mathcal{Z}_0}}{|\mathcal{Z}_0|} h(s_k), \qquad (47)$$

$$\Omega_k = \Omega + (\omega_0 - \Omega) s_k Q(s_k), \qquad (48)$$

where

$$s_k = \frac{\omega_0 - \Omega}{pq_k}.$$

Moreover, the self-consistency equation (37) also simplifies to

$$p = ie^{-i\alpha}\mathcal{W}_0, \qquad (49)$$

where

$$\mathcal{W}_0 = \int_0^{\infty} H_q(q) h\left(\frac{\omega_0 - \Omega}{pq}\right) q \, dq. \qquad (50)$$

Next, we will demonstrate that the parameters $\omega_0$, $\alpha$ and $q_j$ in Eq. (46) can be found from the observed values of $\zeta_j$ and $\Omega_j$. First, we note that

the definitions (36) and (40) imply that $\text{Im}(\mathcal{Z}_0) \leq 0$ and $-\pi \leq \arg \mathcal{Z}_0 \leq 0$. Recalling that $h(s)$ satisfies the quadratic equation $h^2 - 2sh + 1 = 0$ and using relations (47), we write

$$\arg \mathcal{Z}_0 = \phi_* := \text{argmin}_{\phi \in [-\pi, 0]} \sum_{k=1}^{N} \left[ \text{Im} \left( \frac{\zeta_k^2 e^{2i\phi} + 1}{2\zeta_k e^{i\phi}} \right) \right]^2,$$

and

$$s_k = \text{Re} \left( \frac{\zeta_k^2 e^{2i\phi_*} + 1}{2\zeta_k e^{i\phi_*}} \right).$$

Then, the parameter $\Omega$ and the difference $\omega_0 - \Omega$ can be determined by linear regression from equations (48). This allows us to find $pq_k = (\omega_0 - \Omega)/s_k$. Taking into account that formula (50) defines the mean values of the product $qh((\omega_0 - \Omega)/pq)$, we conclude

$$p\mathcal{W}_0 \approx \mathcal{S} := \frac{1}{N} \sum_{k=1}^{N} pq_k h(s_k) = \frac{1}{N} \sum_{k=1}^{N} \frac{\omega_0 - \Omega}{s_k} h(s_k).$$

Substituting this into the self-consistency equation (49), we finally get

$$p = \sqrt{\mathcal{S}}, \quad \alpha = \arg(i\mathcal{S}), \quad \text{and} \quad q_j = \frac{\omega_0 - \Omega}{ps_k}.$$

Note that using the above approach, we can only find the values of $q_k = \sqrt{K} d_k / d_{\text{mean}}$, but we cannot separate them into the coupling strength $K$ and the degree $d_k$. In any case, even this information is enough to decide, which oscillators are more connected and which are less connected, and also what type of distribution $H_q(q)$ characterizes the network.

### Remark about the partial data case

The statistical equilibrium relations (38) and (39) express the relationship between local observables and local parameters. Therefore, they remain valid also for partial observations of $\zeta_k$ and $\Omega_k$. However, to implement the above parameter reconstruction algorithms for models (41) and (46), it is necessary to know the value of a global order parameter such as $\mathcal{Z}_0$ or $\mathcal{W}_0$, see (40) and (50). Fortunately, both of these quantities are defined as averages across the entire network. Therefore, if measurements are only available for a part of the oscillators, their parameters can still be found (albeit with less precision), provided that these oscillators are uniformly distributed across the network. In this case, the global averages should be calculated as averages over the available oscillators.

## Data availability

All data used in this study can be generated using the codes included in the Supplementary Data 1 file.

## Code availability

All numerical codes used in this study are included in the Supplementary Data 1 file. Their description is given in the Description of Supplementary Data 1 file.

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

## Acknowledgements

This study was supported by the Deutsche Forschungsgemeinschaft under grant OM 99/2-3.

## Author contributions

The author (O.E.O.) confirms the sole responsibility for the conception of the study, presented results and manuscript preparation.

## Funding

## Competing interests

The author declares no competing interests.
