## [Transparent Peer Review file · Nature Communications]

Inverse problems for dynamic patterns in coupled oscillator networks: When larger networks are simpler

Corresponding Author: Dr Oleh Omel'chenko

Version 1:

Reviewer comments:

Reviewer #1

(Remarks to the Author)

Report on "Inverse problems for dynamic patterns in coupled oscillator networks: When larger networks are simpler" by Omel'chenko

In this very well written and original contribution the author employs sophisticated (and, to this referee, rather surprising) explicit analytical expressions for certain thermodynamic variables to determine unknown parameters of the dynamical system on a network of many thousand oscillators. As an example to illustrate their method the author chose the challenging problem of chimera states which have attracted much attention in the last 10 years, from physicists, mathematicians and biologists. The author corroborates their theory with excellent numerical results, which are clearly presented, illustrate the efficacy and accuracy of the proposed method and are reproducible given the matlab code, provided by the author.

The methods and general strategy developed here will be applicable to a much wider class of systems (the author actually chose a very challenging one), including power grids. I therefore believe that this work will be of interest to a large part of the scientific community.

All calculations are explicitly given and the numerical code

The author's method to infer unknown parameters from a time series has several advantages over other existing approaches: robustness to (unbiased) noise, non-reliance on equidistantly sampled data, no reliance on knowledge of temporal derivatives (and instead only using smoothing averaging), no requirement on full observations, and no inversion of $N \times N$ matrices (where N is the number of dynamic variables). This makes the newly proposed method stand out against previous methods and makes it a very powerful and widely applicable new methodological framework.

I only have a few comments the author may wish you take on board.

On page 5, the author restricts to symmetric functions $G(x)$. Could the author comment on this choice and also on possible ways to tackle anti-symmetric G or G which do not have any symmetry (are there any chimeras in these situations possible?)?

I was wondering about the number of Fourier modes required to infer the parameters. It is nicely shown that 10 Fourier modes are sufficient to obtain a very good accuracy for the parameters, despite the coupling function $G(x)$ not being accurately represented (as seen in Fig 4, right figure). Does the accuracy grow monotonically with the number of Fourier modes or will the Gibb's phenomenon lead to a degradation?

A brief comment on the computational cost of minimizing the function J would be helpful to know.

The author lists a few works on other approaches to infer unknown parameters, but missed to mention the recent work Smith & Gottwald, Proc Roy Soc A (2025), 481 (2025) 20240813, entitled "Data assimilation for networks of coupled oscillators: Inferring unknown model parameters from partial observations". This method, employing the ensemble Kalman filter and Bayesian inference allows for noisy partial observations (but also smaller networks for which the thermodynamic limit would not be valid), but requires a constant sampling time and knowledge of the coupling function.

Typos:

- * comma after (30 missing
- * page 2: 3rd line in Results: of a chimera state
- * several commata are missing in formulae which are followed by the word “where”

Reviewer #2

(Remarks to the Author)

The author proposed an inverse approach to infer on system parameters and more importantly to reconstruct the coupling functions using the chimera pattern data obtained from the Kuramoto-Battogtokh model with nonlocal coupling, based on observation of the average frequency and the local order parameter. In particular, the author used three different coupling schemes to validate their method. While this approach might be a promising step towards the inverse problem, in general, for the future, with its analytical foundation and numerical confirmation with different coupling functions, I have major concern about the limitations of this method.

1. While the conclusion suggests that the method may have broad applicability, the abstract and introduction do not adequately convey the underlying motivation and physical framework necessary for inverse analysis. A more explicit discussion on the advantages and disadvantages of the proposed approach would be valuable. For example, it is not clear how the reconstruction would perform for systems with more complex connectivity structures—such as small-world networks, and even globally coupled network—rather than the nonlocal topology considered here. It would also be important to clarify whether this method can handle distributed phase lag parameters (α) or heterogeneity in intrinsic parameters such as the natural frequencies. In particular, could partial synchronization arising in classical Kuramoto oscillators with complex connectivity and Lorentzian-distributed frequencies still allow for recovery of the coupling function and frequency set? If not, these limitations should be explicitly acknowledged in the text.
2. The current analysis is limited to phase oscillator networks that exhibit a particular chimera pattern. A broader discussion of how the inverse method might be adapted to other dynamical regimes—such as clustering states, breathing chimeras, or turbulent incoherent states—would strengthen the claim of generality.
3. The study focuses on a chimera configuration with two coherent/incoherent domains. It would be insightful to explore the potential outcomes if multi-headed chimera states were used instead. Would the SER-based reconstruction remain robust? Such an extension could provide valuable insights into the generality and scalability of the proposed approach.
4. It seems, the current work assumes a coupling strength fixed at unity. It would be useful to discuss whether the algorithm could be extended to cases with variable coupling strengths. If so, would the method still reliably infer other parameters under these conditions, or would additional observables/constraints be needed? Addressing this would clarify the flexibility of the method for more realistic scenarios.
5. The reconstruction of coupling functions in nonlinear dynamical systems is a well-established topic. In particular, methods for identifying coefficients in polynomial expansions such as the SINDy framework (Proc. Natl. Acad. Sci. U.S.A. 113, 3932 (2016)) and other related approaches (Proc. Natl. Acad. Sci. U.S.A. 115, 9300 (2018); Phys. Rev. Lett. 130, 117401 (2023); Chaos 29, 103116 (2019)) have been extensively studied. The author should clearly explain the specific novelty of their method compared to these existing techniques. For example, what distinct advantages does the use of statistical equilibrium relations provide such as scalability to very large networks, robustness to measurement noise, or applicability with partial/uneven data that go beyond the capabilities of these prior methods?

Reviewer #3

(Remarks to the Author)

Reviewer #4

(Remarks to the Author)

Version 2:

Reviewer comments:

Reviewer #1

(Remarks to the Author)

The author has addressed all my concerns and I am very happy to recommend publication. Great work!

Just one suggestion the author may take on board if he sees fit: Fig 1 from the response letter could be included in a supplementary file together with a discussion. But this should be left to the author's decision on how this fits.

Reviewer #2

(Remarks to the Author)

The author responded to our questions, hence recommend acceptance for publication.

Reviewer #3

(Remarks to the Author)

Reviewer #4

(Remarks to the Author)

**Response to Reviewers' comments on
"Inverse problems for dynamic patterns
in coupled oscillator networks:
When larger networks are simpler"
by Oleh Omel'chenko**

We thank the Reviewers for their careful reading of our manuscript and for their valuable comments on our results. In the revised version of the manuscript, we have attempted to take all of these comments into account by adding new material and expanding our explanations. We are pleased that both Reviewers recognized the potential of our method and raised questions about how it would work in other situations. We hope that our responses provide a general answer to these questions, and more detail can be obtained from further research. In any case, the aim of our work is to describe a new analytical method by applying it to several paradigmatic examples, rather than an exhaustive description of all the results that can be obtained using it. We hope that the responses below will be found satisfactory by the Reviewers, and that the resubmitted manuscript will be accepted for publication in *Nature Communications*.

Response to Reviewer 1:

In this very well written and original contribution the author employs sophisticated (and, to this referee, rather surprising) explicit analytical expressions for certain thermodynamic variables to determine unknown parameters of the dynamical system on a network of many thousand oscillators. As an example to illustrate their method the author chose the challenging problem of chimera states which have attracted much attention in the last 10 years, from physicists, mathematicians and biologists. The author corroborates their theory with excellent numerical results, which are clearly presented, illustrate the efficacy and accuracy of the proposed method and are reproducible given the matlab code, provided by the author.

The methods and general strategy developed here will be applicable to a much wider class of systems (the author actually chose a very challenging one), including power grids. I therefore believe that this work will be of interest to a large part of the scientific community.

All calculations are explicitly given and the numerical code

The author's method to infer unknown parameters from a time series has several advantages over other existing approaches: robustness to (unbiased) noise, non-reliance on equidistantly sampled data, no reliance on knowledge of temporal derivatives (and instead only using smoothing averaging), no requirement on full observations, and no inversion of $N \times N$ matrices (where N is the number of dynamic variables). This makes the newly proposed method stand out

against previous methods and makes it a very powerful and widely applicable new methodological framework. I only have a few comments the author may wish you take on board.

Response: We thank the Reviewer for this positive assessment of our manuscript.

On page 5, the author restricts to symmetric functions $G(x)$. Could the author comment on this choice and also on possible ways to tackle anti-symmetric G or G which do not have any symmetry (are there any chimeras in these situations possible)?

Response: Our study focuses on *stationary* coherence-incoherence patterns, i.e., patterns that do not change their position on the network and do not change their shape at the macroscopic level. (We have now added a better explanation of this fact to the last two paragraphs of the section “Statistical equilibrium relations (SER)” on page 4.) On the other hand, it is well-known that asymmetric coupling topologies with $G(-x) \neq G(x)$ typically lead to patterns that drift across the network. Such are, for example, travelling chimera states described in [1]. In this case, uniform drift causes the local order parameters of different oscillators (as well as their effective frequencies) to become the same after averaging over a long time interval. This means that all information about the network heterogeneities is lost and cannot be recovered. It is likely that a modified version of our method can be developed that will allow the calculation of local order parameters and effective frequencies in a suitably chosen corotating frame. Then the coupling function $G(x)$ should be sought in the form of a general Fourier series with cosine and sine harmonics $\cos(mx)$ and $\sin(mx)$. However, such a consideration is beyond the scope of the presented study, so we leave it for future research.

I was wondering about the number of Fourier modes required to infer the parameters. It is nicely shown that 10 Fourier modes are sufficient to obtain a very good accuracy for the parameters, despite the coupling function $G(x)$ not being accurately represented (as seen in Fig 4, right figure). Does the accuracy grow monotonically with the number of Fourier modes or will the Gibb’s phenomenon lead to a degradation?

Response: For a given set of local order parameters ζ_k and effective frequencies Ω_k , the L^2 -accuracy of the reconstruction of the coupling function $G(x)$ increases almost monotonically with the number of Fourier modes M , but only up to some threshold value M_{opt} , while for $M > M_{\text{opt}}$ the accuracy starts to deteriorate, see Fig. 1 below. Remarkably, the value M_{opt} is not related to the Gibbs phenomenon, but to the finite accuracy of the statistical equilibrium relations. Since these relations are exact for an infinite number of oscillators N and an infinite averaging time T , we know that $M_{\text{opt}} \rightarrow \infty$ as $N, T \rightarrow \infty$. But a more specific functional dependence of M_{opt} on N and T is difficult, if not impossible, to obtain.

Figure 1: L^2 -accuracy of the reconstructed coupling function $G_{\text{reconstr}}(x)$ relative to the original coupling function $G(x)$. The chimera state for the parameters from Fig. 2 in the manuscript was numerically simulated for system sizes $N = 1024, 2048,$ and 4096 , and the corresponding effective frequencies Ω_k and local order parameters ζ_k were calculated as time-averaged over $T = 2000$. The obtained data were used in our reconstruction algorithm with different M . In each case, the accuracy initially increases with M , but for the number of Fourier harmonics M greater than some M_{opt} starts to deteriorate. The black curve shows the maximum possible accuracy provided by a truncated Fourier series with all spatial modes $\cos(mx)$ where $m \leq M$.

A brief comment on the computational cost of minimizing the function J would be helpful to know.

Response: We followed the Reviewer’s recommendation and added a paragraph discussing the computational cost of our algorithm at the end of the section “Parameter reconstruction algorithm”, see page 6.

The author lists a few works on other approaches to infer unknown parameters, but missed to mention the recent work Smith & Gottwald, Proc Roy Soc A (2025), 481 (2025) 20240813, entitled “Data assimilation for networks of coupled oscillators: Inferring unknown model parameters from partial observations”. This method, employing the ensemble Kalman filter and Bayesian inference allows for noisy partial observations (but also smaller networks for which the thermodynamic limit would not be valid), but requires a constant sampling time and knowledge of the coupling function.

Response: We are very grateful that this recently published work was brought to our attention. We have included it in the bibliography and cited it in the section “Discussion”.

Typos:

** comma after (30 missing)*

** page 2: 3rd line in Results: of a chimera state*

** several commata are missing in formulae which are followed by the word “where”*

Response: We thank the Reviewer for spotting these typos. All they were corrected.

Response to Reviewer 2:

The author proposed an inverse approach to infer on system parameters and more importantly to reconstruct the coupling functions using the chimera pattern data obtained from the Kuramoto-Battogtokh model with nonlocal coupling, based on observation of the average frequency and the local order parameter. In particular, the author used three different coupling schemes to validate their method. While this approach might be a promising step towards the inverse problem, in general, for the future, with its analytical foundation and numerical confirmation with different coupling functions, I have major concern about the limitations of this method.

1. While the conclusion suggests that the method may have broad applicability, the abstract and introduction do not adequately convey the underlying motivation and physical framework necessary for inverse analysis. A more explicit discussion on the advantages and disadvantages of the proposed approach would be valuable. For example, it is not clear how the reconstruction would perform for systems with more complex connectivity structures — such as small-world networks, and even globally coupled network — rather than the nonlocal topology considered here. It would also be important to clarify whether this method can handle distributed phase lag parameters (α) or heterogeneity in intrinsic parameters such as the natural frequencies. In particular, could partial synchronization arising in classical Kuramoto oscillators with complex connectivity and Lorentzian-distributed frequencies still allow for recovery of the coupling function and frequency set? If not, these limitations should be explicitly acknowledged in the text.

Response: We are very grateful to the Reviewer for this extremely important comment. Although our initial intention was to describe the concept of statistical equilibrium relations in the context of chimera states, we now see that other applications should be mentioned in the manuscript for a better understanding of our method. Accordingly, we have added a new section “Other applications” after the section “Methods”. In the new section, we explain how statistical equilibrium relations can be obtained for the classical Kuramoto-Sakaguchi model and for the annealed approximation of the Kuramoto model on a random network. For the first model, we also show how statistical equilibrium relations can be used to find all oscillator frequencies, coupling strength, and phase lag parameter from observed values of local order parameters and effective frequencies. (Importantly, this can be done for arbitrary frequency distributions, not just Lorentzians!) For the second model, representing a Kuramoto-type random network, we

show that statistical equilibrium relations can be used to recover the relative node degrees and their distribution, which can potentially help to identify the type of the network. Finally, we explain that in both of the above cases, the parameter reconstruction algorithm can be applied to partially collected data, provided that the observed oscillators are uniformly distributed across the network.

To reflect the above changes to the text, we have updated the abstract and introduction accordingly.

As for the case of distributed phase lag parameters, it also has a thermodynamic limit description and corresponding statistical equilibrium relations. But these relations contain other functional dependencies, so their analysis in the context of parameter reconstruction will be qualitatively different from the two examples that we have already added to the manuscript in the section “Other applications”. Honestly, we currently do not have a solution to the parameter reconstruction problem in this case and prefer to work on this in a separate paper. Again, we cannot apply our method to all interesting problems at once.

2. The current analysis is limited to phase oscillator networks that exhibit a particular chimera pattern. A broader discussion of how the inverse method might be adapted to other dynamical regimes — such as clustering states, breathing chimeras, or turbulent incoherent states — would strengthen the claim of generality.

Response: The approach developed in our work focuses on stationary coherence-incoherence patterns and we clearly state this in the text. Clustering states, where the oscillator network is divided into several groups with identical dynamical behaviour in each of these groups, are not suitable for parameter reconstruction. And this applies to any parameter reconstruction method, not just ours. Imagine that you observe a two-cluster state in a network of $N \gg 2$ oscillators, then the details about the heterogeneities of the oscillators and the coupling topology in each of the clusters are no longer present in the observed data and therefore cannot be recovered! Similarly, turbulent incoherent states are also a poor source of input data for our method, as we only use time-averaged quantities, and for incoherent states they typically take the same value across the network due to the ergodicity property. As for breathing chimeras, they are non-stationary coherence-incoherence patterns, so the current version of our parameter reconstruction algorithm cannot be applied to them. But, using the analytical approach presented in [2] one can hope to obtain statistical equilibrium relations for this type of patterns as well. However, such a consideration is beyond the scope of the presented study, therefore we leave it for future research. To emphasize this, we have added new text to the sections “Statistical equilibrium relations (SER)”, see page 4, and “Discussion”, see page 10.

3. The study focuses on a chimera configuration with two coherent/incoherent domains. It would be insightful to explore the potential outcomes if multi-headed chimera states were used

instead. Would the SER-based reconstruction remain robust? Such an extension could provide valuable insights into the generality and scalability of the proposed approach.

Response: In general, the multi-headed chimera states found in the Kuramoto-Battogtokh model can be analyzed in the same way as the chimera states with two coherent/incoherent domains. The SER-based parameter reconstruction algorithm remains unchanged. The only problem that may arise is that multi-headed chimera states usually have antiphase adjacent coherent regions, and therefore their global order parameter is almost zero. This circumstance makes the definition of the local order parameters ζ_j given in the Introduction unsuitable. In the new section “Chimera states with zero global order parameter” (see pages 8–9) we explain how this problem can be solved using a Daido order parameter.

4. It seems, the current work assumes a coupling strength fixed at unity. It would be useful to discuss whether the algorithm could be extended to cases with variable coupling strengths. If so, would the method still reliably infer other parameters under these conditions, or would additional observables/constraints be needed? Addressing this would clarify the flexibility of the method for more realistic scenarios.

Response: There is nothing special about the coupling function $G(x)$ being normalized to unity in all our examples. It is simply a tribute to the tradition established in the literature on chimera states. If you examine the rationale for our parameter reconstruction algorithm, you will find that the normalization condition is never used. Therefore, the proposed algorithm works equally well for coupling functions with other normalizations. We added an explanation of this fact at the end of the section “Examples” on page 7.

5. The reconstruction of coupling functions in nonlinear dynamical systems is a well-established topic. In particular, methods for identifying coefficients in polynomial expansions such as the SINDy framework (Proc. Natl. Acad. Sci. U.S.A. 113, 3932 (2016)) and other related approaches (Proc. Natl. Acad. Sci. U.S.A. 115, 9300 (2018); Phys. Rev. Lett. 130, 117401 (2023); Chaos 29, 103116 (2019)) have been extensively studied. The author should clearly explain the specific novelty of their method compared to these existing techniques. For example, what distinct advantages does the use of statistical equilibrium relations provide such as scalability to very large networks, robustness to measurement noise, or applicability with partial/uneven data that go beyond the capabilities of these prior methods?

Response: The SINDy framework is an extremely general method designed to reconstruct an arbitrary n -dimensional dynamical system of the form $\dot{\mathbf{x}} = \mathbf{f}(\mathbf{x})$ from noisy measurement data [3]. But its generality can also be a source of problems. According to its authors, this “method is ill-suited for a large state dimension n , because of the factorial growth of Θ [a library vector consisting of candidate nonlinear functions of \mathbf{x}] in n and because each of the n row equations in Eq. 4 [a system of linear equations that determines the coefficients at different

terms of library functions] requires a separate optimization” [3, p. 3933]. Even if we assume a certain ansatz for $\mathbf{f}(\mathbf{x})$ (which is actually done in the present manuscript), using the SINDy approach we will obtain an $(nm) \times (nm)$ system of linear equations, where m is the number of sampling times. Finding a solution to this system usually requires $O((nm)^3)$ operations, while our SER-based algorithm requires only $O(n^2)$ operations, as explained in the new paragraph of the section “Parameter reconstruction algorithm” on pages 6–7.

The authors of [3] also describe a situation where the curse of dimensionality can be overcome relatively easily: “Fortunately, many high-dimensional systems of interest evolve on a low-dimensional manifold or attractor that is well-approximated using a low-rank basis Ψ ” [3, p. 3933]. Regarding this remark, we note that most partially synchronized states on coupled oscillator networks, including chimera states, are hyperchaotic trajectories with Kaplan-Yorke dimension proportional to the system size n [4]. The low-dimensional nature of the dynamics of such states becomes apparent only in the thermodynamic limit and only when using a probability density representation. This is another feature that is taken into account in our SER-based algorithm, but cannot be simply implemented in the SINDy framework.

The fact that we do not need to calculate time derivatives and can use partial observations is also an advantage of the SER-based algorithm compared to the SINDy approach. But our method is more focused on a specific (albeit quite broad) class of models and must be tuned to each model individually.

A more specific advantage of our method is explained in the new section “Calculations with protophases” on pages 9–10. We show that the effective frequencies and local order parameters required for our parameter reconstruction algorithm can be easily calculated even in the case where, instead of the physical phases $\theta_j(t)$, we observe their transformed values $\phi_j(t) = \Phi(\theta_j(t))$ with an unknown transformation function $\Phi(\theta)$. This situation is typical in real-world applications, where the physical phase is not measured directly, but is reconstructed from a protophase obtained from an oscillatory signal [5].

Response to Reviewer 3:

Response: All comments from the reports of Reviewer 1 and Reviewer 2 were very useful to us and helped to improve the presentation and clarity of our results. Therefore, we would like to thank Reviewer 3 for his/her contribution to one of the above reports.

Response to Reviewer 4:

Response: All comments from the reports of Reviewer 1 and Reviewer 2 were very useful to us and helped to improve the presentation and clarity of our results. Therefore, we would like to thank Reviewer 4 for his/her contribution to one of the above reports.

References:

- [1] O. E. Omel'chenko, Travelling chimera states in systems of phase oscillators with asymmetric nonlocal coupling, *Nonlinearity* **33**, 611–642 (2020).
- [2] O. E. Omel'chenko, Mathematical framework for breathing chimera states, *J. Nonlinear Sci.* **32**, 22 (2022).
- [3] S. L. Brunton, J. L. Proctor, and J. N. Kutz, Discovering governing equations from data by sparse identification of nonlinear dynamical systems, *Proc. Natl. Acad. Sci. USA* **113**, 3932–3937 (2016).
- [4] M. Wolfrum, O. E. Omel'chenko, S. Yanchuk, and Y. L. Maistrenko, Spectral properties of chimera states, *Chaos* **21**, 013112 (2011).
- [5] B. Kralemann, L. Cimonieriu, M. Rosenblum, A. Pikovsky, and R. Mrowka, Phase dynamics of coupled oscillators reconstructed from data, *Phys. Rev. E* **77**, 066205 (2008).

**Response to Reviewers' comments on
"Inverse problems for dynamic patterns
in coupled oscillator networks:
When larger networks are simpler"
by Oleh Omel'chenko**

We thank the Reviewers for all their comments (from this and previous reports), which greatly helped us to improve the quality of our work. We hope that with the revisions made, the manuscript will finally be accepted for publication in *Nature Communications*.

Response to Reviewer 1:

The author has addressed all my concerns and I am very happy to recommend publication. Great work!

Just one suggestion the author may take on board if he sees fit: Fig. 1 from the response letter could be included in a supplementary file together with a discussion. But this should be left to the author's decision on how this fits.

Response: We thank the Reviewer for this positive assessment of our work. We followed his/her suggestion and added Fig. 1 from the previous response letter along with its discussion to the Supplementary Information file (see Supplementary Note 1). Accordingly, we added two new sentences referring to this note in the main text after Eq. (14).

Response to Reviewer 2:

The author responded to our questions, hence recommend acceptance for publication.

Response: We are glad we were able to answer all of the Reviewer's questions, which prompted us to describe additional aspects of our method.

Response to Reviewer 3:

Response: All comments from the reports of Reviewer 1 and Reviewer 2 were very useful to us and helped to improve the presentation and clarity of our results. Once again, we would like to thank Reviewer 3 for his/her contribution to one of the above reports.

Response to Reviewer 4:

I co-reviewed this manuscript with one of the reviewers who provided the listed reports. This is part of the NatureCommunications initiative to facilitate training in peer review and to provide appropriate recognition for Early CareerResearchers who co-review manuscripts.

Response: All comments from the reports of Reviewer 1 and Reviewer 2 were very useful to us and helped to improve the presentation and clarity of our results. Once again, we would like to thank Reviewer 4 for his/her contribution to one of the above reports.